# Silver Nanoparticles Alter Cell Viability Ex Vivo and in Vitro and Induce Proinflammatory Effects in Human Lung Fibroblasts

**DOI:** 10.3390/nano10091868

**Published:** 2020-09-18

**Authors:** Anna Löfdahl, Andreas Jern, Samuel Flyman, Monica Kåredal, Hanna L Karlsson, Anna-Karin Larsson-Callerfelt

**Affiliations:** 1Unit of Lung Biology, Department of Experimental Medical Sciences, Lund University, SE-221 00 Lund, Sweden; Anna.Lofdahl@med.lu.se (A.L.); Andreas.Jern@med.lu.se (A.J.); sflymanvii@gmail.com (S.F.); 2Division of Occupational and Environmental Medicine, Lund University, SE-221 00 Lund, Sweden; Monika.Karedal@med.lu.se; 3Unit of Biochemical Toxicology, Institute of Environmental Medicine, Karolinska Institutet, SE-171 76 Stockholm, Sweden; Hanna.l.karlsson@ki.se

**Keywords:** silver nanoparticles, human lung fibroblasts, precision-cut lung slices, toxicity, extracellular matrix, procollagen, growth factors, cytokines

## Abstract

Silver nanoparticles (AgNPs) are commonly used in commercial and medical applications. However, AgNPs may induce toxicity, extracellular matrix (ECM) changes and inflammatory responses. Fibroblasts are key players in remodeling processes and major producers of the ECM. The aims of this study were to explore the effect of AgNPs on cell viability, both ex vivo in murine precision cut lung slices (PCLS) and in vitro in human lung fibroblasts (HFL-1), and immunomodulatory responses in fibroblasts. PCLS and HFL-1 were exposed to AgNPs with different sizes, 10 nm and 75 nm, at concentrations 2 µg/mL and 10 μg/mL. Changes in synthesis of ECM proteins, growth factors and cytokines were analyzed in HFL-1. Ag10 and Ag75 affected cell viability, with significantly reduced metabolic activities at 10 μg/mL in both PCLS and HFL-1 after 48 h. AgNPs significantly increased procollagen I synthesis and release of IL-8, prostaglandin E_2_, RANTES and eotaxin, whereas reduced IL-6 release was observed in HFL-1 after 72 h. Our data indicate toxic effects of AgNP exposure on cell viability ex vivo and in vitro with altered procollagen and proinflammatory cytokine secretion in fibroblasts over time. Hence, careful characterizations of AgNPs are of importance, and future studies should include timepoints beyond 24 h.

## 1. Introduction

Silver nanoparticles (AgNPs) are widely used in commercial and medical applications, such as in products for disinfectants, water purification and band aids to improve wound healing, due to their antibacterial properties and characteristic features [1]. AgNPs are particles with sizes between 1 nm and 100 nm that may possess different characteristics related to size, shape, and chemical composition [2]. Nanoparticles of all types, including silver, may also be used to carry other particles for delivery into specific cells, such as in treatments for cancer [3] or chronic lung diseases [4]. The commercial use of AgNPs increases the risk that the particles are spread throughout the environment causing adverse health effects [1]. Studies have shown that AgNPs may have harmful effects on various cell types [2]. AgNPs have been shown to directly interact with the extracellular matrix (ECM) and improve its stability in general [5]. In a previous in vitro study, long-term exposure of lung epithelial cells to AgNPs induced cytotoxicity, epithelial–mesenchymal transition (EMT) and increased collagen deposition [6,7], thus implicating a pro-fibrotic response by the AgNPs. Upon damages of the airway epithelial cell layer due to chronic lung disease or chronic exposure of inhaled particles, cells beneath the epithelial basement membrane will become exposed to nanoparticles, triggering an inflammatory response that may promote locally enhanced levels of pro-inflammatory cytokines. Fibroblasts are mesenchymal cells, crucial for maintaining homeostasis of ECM proteins, such as collagens and proteoglycans, components that together act as important reservoirs for cytokines and growth factors [8,9]. The ECM provides steady, elastic and durable walls of the alveoli, which are necessary for breathing efficiently and creating low resistance for gas exchange. Fibroblasts are also contributing in inflammation processes and respond to secreted growth factors, chemokines and cytokines, which stimulate a pro-fibrotic fibroblast phenotype that promotes tissue remodeling [10]. Fibroblasts are involved in the chronic development of lung fibrosis in several severe lung diseases [9,11,12]. However, studies on the effect of AgNPs in fibroblasts have primarily been performed during 24 h of particle exposure, where a cytotoxic effect was observed with smaller AgNPs [13,14]. In the present study, which is partly a follow up study to Gliga et al., [6,7], we have used AgNPs with different sizes (10 nm and 75 nm) and concentrations (2 μg/mL and 10 μg/mL) up to 72 h to study potential changes on cell viability, ECM synthesis and cytokine profile in human lung fibroblasts and in murine lung tissue slices ex vivo. Our obtained data indicate that AgNPs possess immunomodulatory and proinflammatory effects and are able to stimulate an increased synthesis of procollagen I in human lung fibroblasts. Importantly, AgNPs were also shown to have dose-dependent effects on cell viability observed both in vitro and ex vivo over time, findings that may be of relevance for future biomedical applications. 

## 2. Materials and Methods 

Human lung fibroblast cultures: Human fetal lung fibroblasts (HFL-1) (CCL-153, ATCC, Manassas, VA, USA) were cultured in Dulbecco’s modified Eagle medium (DMEM, Sigma Aldrich, St Louis, MO, USA) supplemented with 10% fetal clone serum (FCIII, Thermo Scientific, Waltham, MA, USA), 1% L-glutamine, 0.5% gentamicin and 5 μg/mL amphotericin B (all from Gibco BRL, Paisley, UK) at 37 °C, 10% CO_2_. HFL-1 was used in passage 16–21. DMEM supplemented with 0.4% FCIII serum was used throughout all experimental conditions to avoid interactions with factors in the serum.

Precision cut lung slices (PCLS): C57/Bl6 mice (female, >10 weeks) (Scanbur research A/S, Karlslunde, Denmark) were sacrificed with an i.p. injection of pentobarbital (0.3 mL of 60 mg/mL). Lungs were filled with pre-warmed low-melting agarose (Sigma Aldrich) solution (0.75% agarose) via a fixed positioned needle inserted into the trachea as previously described [11]. After 10 min on ice, heart and lungs were removed and put in a tube with minimal essential medium (MEM) supplemented with amino acids (100×), vitamins (100×), sodium pyruvate 1% and glutamine (1%) and PEST (1%). Tubes were placed on ice to allow the agarose to further solidify. Cores of lung tissue were manually cut and then further sectioned into 300 µm thick lung slices using a Krumdieck tissue slicer (Alabama Research and Development, Munford, AL, USA). Slices were incubated at 37 °C, 5% CO_2_ overnight, with repeated medium change during the first 4 h. Animal experiments were approved by the local ethics committee for animal research in Malmö/Lund, Sweden (Dnr M103/14).

Ag nanoparticles and characterization: AgNPs in two different sizes; 10 nm Citrate BioPure™ Silver and 75 nm Citrate BioPure™ Silver, were purchased from NanoComposix, Inc. (San Diego, CA, USA) in the form of stock dispersions (1 mg/mL) in aqueous 2 mM citrate. The AgNPs were previously evaluated for their primary size by transmission electron microscopy (TEM) and for their agglomeration in cell medium by photon cross correlation spectroscopy (PCCS) and ultraviolet-visible (UV-vis) spectroscopy. TEM imaging confirmed their primary size, and PCCS/UV-vis analysis showed agglomeration and sedimentation in cell medium over time [6,7,15]. The AgNPs were negative for endotoxin contamination in the limulus amebocyte lysate (LAL) test [15].

Stimulations with Ag nanoparticles: HFL-1 and PCLS were exposed to AgNPs with different sizes, 10 nm (Ag10) and 75 nm (Ag75), and concentrations 2 μg/mL and 10 μg/mL during 24 h, 48 h or 72 h. Prior to cell exposure, dispersions (1000× and 500× dilutions of Ag10 and Ag75) were prepared fresh in fully supplemented cell culture medium containing 0.4% FCIII. Cells and PCLS that were exposed to cell culture medium containing 0.4% FCIII without any AgNPs were regarded as the untreated controls for the experiments. 

Analysis of cytotoxicity and metabolic activity: Cells were cultured in 24-well cell culture plates and exposed to AgNPs for 24 h or 48 h at 37 °C, 10% CO_2_. Cell medium was extracted and cytotoxicity was analyzed for Lactate Dehydrogenase (LDH) release with a LDH Activity Assay Kit (Roche, Cat. No. 11 644 739 001, Basel, Switzerland) according to the manufacturer’s instructions. Cells treated with 1% Triton-X100 were used as a positive control for cell death. Cell metabolism was measured by incubating the HFL-1 cells and PCLS with Tetrazolium salt (WST-1) (Roche, 11 644 807 001) with subsequent absorbance measurements in the collected cell medium, all according to the manufacturer’s instructions.

Quantification of cell amount: Cell amount was determined at 24 h and 48 h, as previously described [10]. In short, HFL-1 cells were plated in 96-well culture plates (Cellstar, Monroe, NC, USA) for 6 h and then exposed to AgNPs for 24 h and 48 h. Cells were fixed in 1% glutaraldehyde (Sigma-Aldrich, St. Louis, MO, USA) for 30 min and stained with 0.1% crystal violet (Sigma-Aldrich, St. Louis, MO, USA) for 30 min. Excess staining solution was washed away and cells were permeabilized overnight with 1% Triton X100 at 4 °C (Merck, Darmstadt, Germany). Changes in cell amount were quantified with a spectrophotometer plate reader, measuring absorbance at 595 nm in collected cell suspension.

Transmission Electron microscopy: HFL-1 and PCLS exposed overnight to AgNPs were fixed in buffer containing 0.1 M Sorensen’s phosphate buffer pH 7.4, 1.5% formaldehyde and 1.5% glutaraldehyde at RT for 2 h for TEM. After fixation, the samples were washed twice in 0.1 M Sorensen’s phosphate buffer pH 7.4 before being dehydrated in a graded series of ethanol (50%, 70%, 80%, 90% and twice in 100%). Samples were critical point dried before being mounted and examined in a Jeol JSM-7800F FEG-SEM at Lund University Bioimaging Center (LBIC).

Analysis of proteoglycans and collagens: Proteoglycan production was determined as previously described [10]. Briefly, HFL-1 were cultured in 6-well culture plates until confluency. Cells were incubated in low sulphate DMEM (Gibco BRL, Paisley, UK) and exposed to AgNPs or transforming growth factor (TGF-β1) Sigma Aldrich) 10 ng/mL (regarded as positive control) in ^35^S–containing DMEM supplemented with 0.4% FCIII, 1% glutamine and PEST 1% for 72 h. Proteoglycans in the cell supernatant were purified by ion exchanger DEAE52 and ^35^S-sulphate incorporation was quantified on a scintillation counter (Wallac; Perkin Ellmer, Boston, MA, USA). Total collagen synthesis was performed using the SircolTM Soluble Collagen assay kit (Biocolor, Carrickfergus, UK) by analyzing the total amount of collagen secreted in the cell supernatant, performed according to the manufacturer’s instructions. The collagen detection limit was 1 μg collagen. Procollagen I α1 synthesis was analyzed in the cell supernatant by ELISA (R&D Systems, Abingdon, UK), following the manufacturer’s instructions. The detection limit for procollagen I α1 was 15.6 pg/mL.

Analysis of growth factors and cytokines: Cell expression of the following interleukins (IL), chemokines and growth factors: IL-6, IL-8, (fibroblast growth factor (FGF-basic, also known as FGF-2), human growth factor (HGF), monokine induced by gamma (MIG), monocyte chemoattractant protein 1 (MCP-1), eotaxin, macrophage inflammatory protein (MIP)-1β, MIP-1α, β-nerve growth factor (NGF), regulated upon activation, normal T cell expressed and secreted (RANTES), platelet derived growth factor (PDGF)-BB and vascular endothelial growth factor (VEGF)-A were determined in cell supernatants collected 72 h post exposure. Analysis was performed with a multiplexed immunoassay on a Bio-Plex 200 Luminex instrument by mixing single-plex assays for the 13 above mentioned growth factors and cytokines (Bio-Rad, Hercules, CA, USA). Undiluted samples were analyzed and sample preparations were performed according to the manufacturer’s instructions. The baseline level of each cytokine measured in cell culture medium containing 0.4% FCIII serum (without any cells) was subtracted from the measured sample concentrations. The calibration curves were fitted using a five-point regression model and results were evaluated in the Bio-Plex Manager Software 6.0 (Bio-Rad). The limit of detection (LOD) for IL-6: 0.3 pg/mL, IL-8: 0.8 pg/mL, FGF-basic: 8.1 pg/mL, HGF: 6.7 pg/mL, MIG: 1.7 pg/mL, MCP-1: 0.7 pg/mL, eotaxin 0.1 pg/mL, MIP-1β: 0.07 pg/mL, MIP-1α: 0.1 pg/mL, β-NGF: 0.2 pg/mL, RANTES: 0.8 pg/mL, PDGF-BB: 2.6 pg/mL and VEGF: 20 pg/mL. Prostaglandin E_2_ (PGE_2_) synthesis was measured in the fibroblast medium by a commercially available enzyme immune assay (EIA) kit (Cayman Chemicals, Limhamn, Sweden). LOD for PGE_2_ was 7.8 pg/mL. All samples were normalized to individual protein amount.

Data presentation and statistical analysis: Data for HFL-1 are presented as mean +/− SEM with n = number of individually performed experiments. For PCLS, n = number of animals used in experiment. Changes in cell metabolism were expressed as % of control and analyzed with one-sample *t*-test. The non-parametric Wilcoxon test was used to compare statistical differences between two groups. Two-way repeated ANOVA measurement on ranks followed by the non-parametric post hoc test Dunn were used to compare differences in cell behavior in response to AgNP exposure. Differences were considered to be statistical significant at *p* < 0.05. All analyses were performed using the software program GraphPad Prism 7.0 (San Diego, CA, USA). 

## 3. Results

### 3.1. AgNP Exposure Affects Viability in Murine PCLS and Human Lung Fibroblasts

Both 2 μg/mL and 10 μg/mL of Ag10 significantly reduced metabolic activity in murine PCLS after 24 h (*p* < 0.01; Figure 1A) and 48 h (*p* < 0.01 and *p* < 0.05; Figure 1B) of exposure. Ag75 did not affect viability at 2 μg/mL after 24 h (Figure 1A), whereas 48 h exposure reduced metabolic activity (*p* > 0.05; Figure 1B). There was a strong tendency (*p* = 0.055) that 10 μg/mL of Ag75 reduced viability in PCLS already after 24 h, whereas a significant decrease in metabolic activity was observed after 48 h (*p* < 0.0001; Figure 1B). Similar results were obtained in lung fibroblasts after exposure for AgNPs. Furthermore, 10 μg/mL of Ag10 reduced metabolic activity in HFL-1 after both 24 h (*p* < 0.001; Figure 1C) and 48 h (*p* < 0.001; Figure 1d), whereas Ag10 at 2 μg/mL showed increased metabolic activity after 48 h (*p* < 0.01; Figure 1D). Ag75 did not affect metabolic activity at 2 μg/mL at neither 24 h nor 48 h (Figure 1C,D), whereas 10 μg/mL of Ag75 resulted in a significant reduction in metabolic activity (*p* < 0.0001) in HFL-1 after 48 h (Figure 1D). 

When examining LDH release, Ag10 and Ag75 did in general not show any cytotoxic effects after 24 h (Figure 2A). However, Ag10 at 10 μg/mL significantly increased the release of LDH in HFL-1, indicating cytotoxic effects after 48 h (*p* < 0.05; Figure 2B). Moreover, 10 μg/mL of Ag75 at 48 h also induced cytotoxic effects after 48 h (*p* < 0.001, Figure 2B). However, Ag75 has previously been shown to interfere with the LDH measurements as notified by Gliga et al. [7]. To further verify the effect of AgNP on cell viability, we measured the amount of cells after 24 h and 48 h. In the untreated cells, 48 h of culture generated a significantly increased amount of cells as compared to 24 h (*p* < 0.001, Figure 2C,D). Ag75 at 10 μg/mL increased the amount of cells, indicating either an increased proliferation or an elevated cell viability after 24 h of exposure (*p* < 0.01; Figure 2C). However, 48 h of exposure to either 10 μg/mL of Ag10 (*p* < 0.05) or 10 μg/mL of Ag75 (*p* < 0.01) significantly reduced the numbers of cells compared to control (Figure 2D). These data could indicate reduced proliferative rate or viability as in line with the obtained results on decreased metabolic activity and increased LDH release after 48 h of AgNP exposure. 

TEM images of AgNPs showed that Ag10 and Ag75 were localized in areas where the gas exchange occurs between alveoli and capillaries in the parenchyma in PCLS (Figure 3C,D). Ag10 and Ag75 were localized in vacuoles or in lysosomal structures in HFL-1 (Figure 3E–H). The mitochondria appeared to have altered appearance in close proximity to the AgNPs (Figure 3H), which may explain our data on reduced metabolic activity since the WST-1 assay indirectly measures mitochondrial activity. 

### 3.2. AgNPs Increased Procollagen I Synthesis in Human Lung Fibroblasts

Ag10 (2 μg/mL) reduced synthesis of proteoglycans (*p* < 0.05; Figure 4A). AgNPs did otherwise not have any significant effects on synthesis of either proteoglycans or total amount of collagens (Figure 4B) at 72 h, whereas the positive control TGF-β1 significantly induced synthesis of proteoglycans (*p* < 0.01; Figure 4A) and total amount of collagens (*p* < 0.05; Figure 4B). However, individual analysis of procollagen I indicated that the release of procollagen I was significantly increased by all the AgNPs and by TGF-β1 after 72 h of exposure (Figure 4C). 

### 3.3. AgNPs Possess Immunomodulatory Effects on Human Lung Fibroblasts

AgNPs triggered proinflammatory and immunomodulatory effects in AgNP exposed HFL-1 cells (Figure 5). Ag10 at 2 μg/mL (*p* < 0.01) and 10 μg/mL (*p* < 0.05) significantly reduced IL-6 release (Figure 5A), whereas Ag10 at 10 μg/mL significantly increased the release of IL-8 (*p* < 0.01; Figure 5B) and PGE_2_ (*p* < 0.05; Figure 5C), and reduced eotaxin (*p* < 0.05; Figure 5G). Moreover, 2 μg/mL of Ag75 significantly reduced IL-6 release (*p* < 0.01; Figure 5A) and MCP-1 (*p* < 0.05, Figure 5E). In addition, 10 μg/mL of Ag75 significantly induced an increased release of IL-8 (*p* < 0.001; Figure 5B) and RANTES (*p* < 0.05; Figure 5D). TGF-β significantly reduced the release of RANTES (*p* < 0.001; Figure 5D), MCP-1 (*p* < 0.001; Figure 5E), MIG (*p* < 0.001; Figure 5F) and eotaxin (*p* < 0.0001; Figure 5G). Protein levels of MIP-1β and MIP-1α were not detectable in the cell supernatants. 

AgNPs did not have any major effects on the different growth factors (Figure 6). However, there was a strong tendency that Ag75, 2 μg/mL (*p* = 0.054) and 10 μg/mL (*p* = 0.064), induced an increased release of PDGF-BB (Figure 6A). Moreover, 10 μg/mL of Ag75 significantly reduced the levels of secreted HGF (*p* < 0.01; Figure 6B), and there was also a strong tendency towards reduced HGF release after exposure to Ag10 10 μg/mL (*p* < 0.054; Figure 6B). TGF-β1 significantly reduced PDGF-bb (*p* < 0.001; Figure 6A), HGF (*p* < 0.0001, Figure 6B) and FGF-basic (*p* < 0.01; Figure 6C) and increased the release of VEGF (*p* < 0.001; Figure 6E). 

## 4. Discussion

In the current study, we evaluated the effect of AgNPs both on tissue level using murine PCLS and on cellular level using human lung fibroblasts. Our obtained data indicate that AgNPs affect cell viability in a concentration-dependent manner, detected both ex vivo and in vitro. Moreover, 10 μg/mL of either Ag10 or Ag75 significantly reduced metabolic activity after 48 h in both HFL-1 and PCLS. The PCLS technique is a well-known methodology for toxicity and pharmacological evaluations [16,17,18,19,20]. The advantages of this technology are that the lung slices are viable for several days and have a maintained tissue integrity with intact pulmonary airways, vessels and surrounding lung parenchyma with alveoli, and it is a model system that in more detail mimics in vivo toxicity in comparison to standard cell culture conditions. From one murine lung, several sequential lung slices are obtained which enables inter- and intra-comparisons and significantly reduces the number of animals used in the experimental set up [11,19]. The effect of AgNPs on cell viability was further evaluated and confirmed in human cells, with increased cytotoxicity and reduced cell amount after 48 h in HFL-1. Several studies have investigated cytotoxic effects of AgNPs and showed that 24 h of exposure with smaller AgNP size (<10 nm) induced more cytotoxic effects than larger AgNPs seen in different cellular in vitro systems, such as human dermal and pulmonary fibroblasts [21], human bronchial epithelial cells [7], the human neural precursor cell line [22] and human macrophage-like THP cells [15], which are in line with our results. However, other studies could not identify any effects of AgNPs on cell viability in fibroblasts [23], and only slight cytotoxic responses were induced in rat PCLS after 24 h of exposure to polyvinylpyrrolidone (PVP)-coated AgNPs (20–30 μg/mL) with a size of 70 nm [24]. However, these AgNPs appeared to remain on the lung slice surface area [24] in contrast to our study, where the AgNPs were localized within the slices, as visualized by TEM. These contradictory results could depend on several factors such as how AgNPs were produced, e.g., if or what type of particle coating was used, the size and concentration of the AgNPs and selected exposure times. However, few studies have investigated timepoints that extend 24 h of particle exposure and none have, to our knowledge, looked at the effect of longer AgNP exposure on lung tissue ex vivo. In this study, we evaluated the effects of AgNPs on cell viability after both 24 h and 48 h. Importantly, we could detect reduced metabolic activity both ex vivo and in vitro already after 24 h with 10 μg/mL of Ag10, but the effect was even more pronounced after 48 h. Similar effects were also obtained with 10 μg/mL of Ag75 after 48 h, indicating that the concentration and not the size of the AgNPs triggers to a greater extent the cellular response over time. Several studies points toward the notion that AgNPs and intracellular released Ag ions may affect cell viability through different routes, such as genotoxicity with DNA damages or mitochondrial disruption followed by induction of oxidative stress and reactive oxygen species (ROS) formation, which in turn induce cytotoxicity and cell apoptosis [7,25,26,27,28]. In our study, TEM imaging of fibroblasts demonstrated that mitochondria in close proximity to AgNPs had structural changes that may indicate dysfunctional mitochondria, which could further explain and support our findings regarding the pronounced reduction in metabolic activity, as seen in AgNP-exposed HLF-1 and PCLS. However, further studies are warranted to follow up these observations.

AgNPs have also been shown to interact with the ECM [5] and may stabilize the ECM structures in scaffold constructs [29]. AgNPs are known to regulate collagen deposition, and AgNP composite nanofibers have been shown to increase collagen production during wound healing [30,31] and may thereby indirectly induce tissue remodeling. Larger AgNPs (>20 nm) actively bind to charged components in the ECM, whereas smaller particles may pass through the ECM network and induce more toxic responses [5]. In a previous long-term study (6 weeks) in human bronchial epithelial cells, it was observed that AgNPs induced pro-fibrotic responses at both gene and protein level, such as increased collagen deposition and epithelial-mesenchymal transition [6]. In the present study, we could not detect any major changes in ECM synthesis of either proteoglycans (except at 72 h exposure of Ag10, 2 µg/mL) or total amount of collagens by fibroblasts, which are vital components in ECM homeostasis [8]. However, we analyzed the total amount of soluble proteoglycans and collagens which could be discriminated if changes were present on individual ECM proteins. When we analyzed the effect of AgNPs on one of the most important procollagens for collagen fibrillization, we could identify that the AgNPs induced an increased amount of procollagen I α1 in human lung fibroblasts after 72 h. The growth factor TGF-β1 which is known to induce powerful pro-fibrotic responses in fibroblasts [11] also significantly increased procollagen I synthesis. Further studies of the ECM proteome with advanced mass spectrometry analysis are warranted to more closely follow potential interactions of AgNPs with individual ECM proteins. The ECM also provides storage for different growth factors and cytokines, where the interplay of AgNPs with the surrounding ECM may be actively involved in remodeling processes. In the present study, we did not show any major effects of AgNP exposure on the secretion of growth factors in lung fibroblasts. However, higher concentrations of both Ag75 and Ag10, reduced HGF levels and exposure for Ag75 showed strong tendencies toward increased PDGF-BB levels, whereas TGF-β1 significantly reduced both PDGF-bb, HGF, and FGF-basic. Collectively, these data indicate that AgNPs and TGF-β1 may induce pro-fibrotic responses through different signaling pathways in human lung fibroblasts.

We also investigated the effect of AgNPs on different cytokines secreted by the fibroblasts after 72 h of exposure. Others have shown that AgNPs may induce both pro- and anti-inflammatory responses in different in vitro systems, such as upregulation of IL-8 secretion by macrophage-like cells [32] and by neutrophils [33]. AgNPs may also interfere with toll-like receptor signaling in cellular innate immunity responses [15]. We demonstrated that IL-8 and RANTES, which belong to the same cytokine family, were upregulated by Ag75 and to same extent by Ag10. RANTES, also known as CCL5, is chemotactic for leukocytes, such as T cells, eosinophils and basophils, whereas IL-8 is a well-known chemokine, attracting neutrophils to inflammatory sites. IL-8 may also promote angiogenesis [5]. IL-8 is also reported to be upregulated during oxidative stress, linking to the reduced metabolic activity induced by 10 μg/mL of Ag10 and Ag75 in HFL-1 seen in this study. In the present study, 2 μg/mL of Ag75 reduced secretion of MCP-1, which is a chemoattractant for monocytes, and 2 μg/mL of Ag10 and Ag75 reduced secretion of IL-6. IL-6 is known as an important pro-inflammatory cytokine that recruits immune cells. IL-6 is also recognized as a myokine, important for repair processes in the tissue [34]. 

In vivo studies on rats have shown that intratracheal installation of 75–150 µg of PVP-coated Ag50 caused a reversible inflammation, and that 300 µg caused DNA damage and neutrophilic granulocytes [35]. Our findings imply that fibroblasts exposed by AgNPs become activated and recruit immune cells as a proinflammatory response. Of note, previous studies in dermal fibroblasts have shown that PDGF-BB, MCP-1, eotaxin and RANTES may increase type I collagen production [36], findings that may explain the upregulation of procollagen I as a response to the enhanced release of PDGF-BB and RANTES observed in the present study. The proinflammatory enzyme cyclooxygenase 2 (COX-2) has been shown to be induced by dermal fibroblasts after 24 h of exposure to AgNPs (10 nm, 25 ppm) [14]. To follow up these results, we examined the down-stream mediator PGE_2_, which is generated via COX, and 10 μg/mL of Ag10 significantly induced PGE_2_ secretion from the fibroblasts.

Although we observed toxic effects of AgNPs both ex vivo and in vitro, this study has some limitations. Our in vitro experiments were performed with a primary fetal cell line (HFL-1) and not using primary cells derived from healthy individuals. The promising results from this study showing a shift in cytokine profile warrants further investigation by co-culturing and investigating fibroblast interactions with inflammatory cells. In order to further increase clinical translatability, human lung slices can also be examined in addition to the murine PCLS, however, this was not feasible in this experimental set up.

In conclusion, our data indicate that immunomodulatory responses are induced by both sizes (Ag10 and Ag75) already at a relatively low dose (2 µg/mL) in human lung fibroblasts. AgNPs significantly affected cell viability both ex vivo in lung tissue slices and in vitro in HFL-1, with significantly reduced viability after 48 h of exposure to higher concentrations (10 μg/mL) of both Ag10 and Ag75. To translate our observed findings to the in vivo situation, previous studies have indicated that daily work exposure of AgNPs over a period of 74 working weeks would correspond to a total cellular deposition equal to the concentration of 10 μg/mL [7]. The used concentrations in this experimental set up are quite high but within the range to be reached after exposure during an acute working accident or after many years of daily exposure. The ECM and cytokine profiles in human lung fibroblasts were clearly influenced by the exposure to AgNPs, promoting increased procollagen I synthesis and a proinflammatory cascade of cytokine release. Hence, careful characterization of AgNPs is of importance, including timepoints beyond 24 h for future applications in nanomaterials, and being aware of toxic effects as well as the ability to influence the ECM and the immune responses over time.

## Figures and Tables

**Figure 1 nanomaterials-10-01868-f001:**
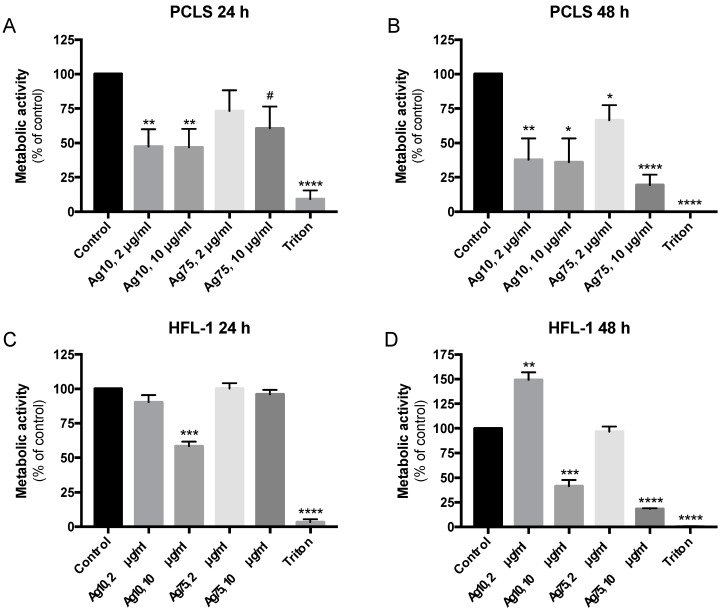
Effect of AgNP exposure on viability measured as changes in metabolic activity by Tetrazolium salt (WST-1) assay in murine precision cut lung slices (PCLS; n = 6) after (**A**) 24 h and (**B**) 48 h and in human lung fibroblasts (HFL-1) (n = 5) after (**C**) 24 h and (**D**) 48 h compared to untreated cells or PCLS (control) and the positive control for cell death, 1% Triton-X100 (Triton). Absorbance of WST-1 measurements in PCLS were adjusted for wet weight for each lung slice sample. Data are presented as % metabolic activity of the control with mean +/− SEM, and statistical analyses are compared to untreated controls. * *p* < 0.05, ** *p* < 0.01, *** *p* < 0.001, **** *p* < 0.0001. # Trend (*p* = 0.055).

**Figure 2 nanomaterials-10-01868-f002:**
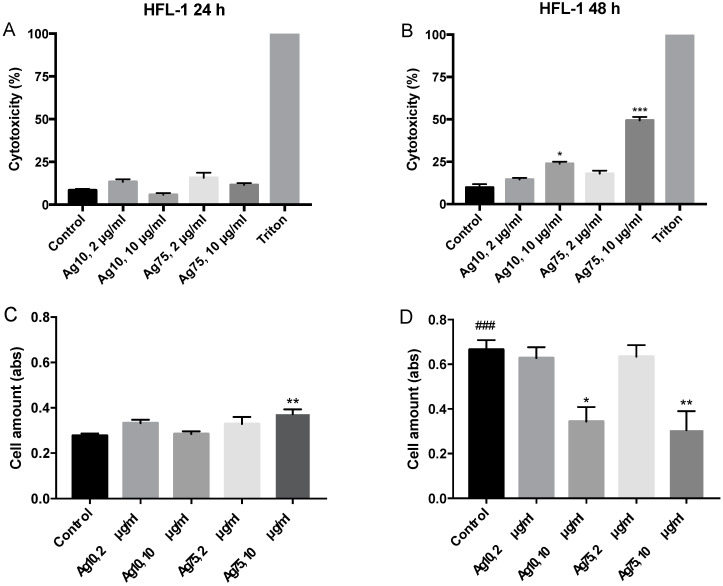
Changes in cytotoxicity measured by Lactate Dehydrogenase (LDH) release by HFL-1 (n = 5) after silver nanoparticle (AgNP) exposure for (**A**) 24 h and (**B**) 48 h compared to control and expressed in % to the positive control (1% Triton-X100 (100% cytotoxicity)). Effect of AgNP exposure on cell amount of HFL-1 after (**C**) 24 h and (**D**) 48 h (n = 5). Data are presented as mean +/− SEM, and statistical analyses are compared to untreated controls. * *p* < 0.05, ** *p* < 0.01, *** *p* < 0.001, ^###^
*p* < 0.001.

**Figure 3 nanomaterials-10-01868-f003:**
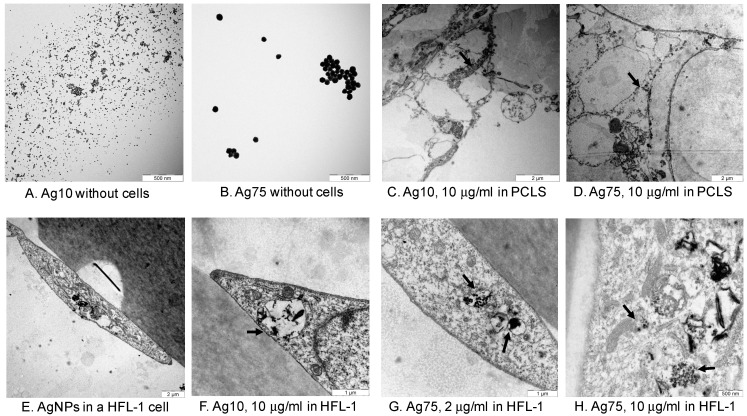
TEM images of Ag10 and Ag75 without any cells (**A**,**B**), in parenchyma in murine PCLS (**C**,**D**) and in HFL-1 (**E**–**H**). Arrows indicate AgNPs localized in alveolar epithelium in PCLS (**C**,**D**) and in vacuoles or lysosomal structures in HFL-1 (Figure 3**F**–**H**) and the altered mitochondria appearance in close proximity to AgNPs (Figure 3**H**).

**Figure 4 nanomaterials-10-01868-f004:**
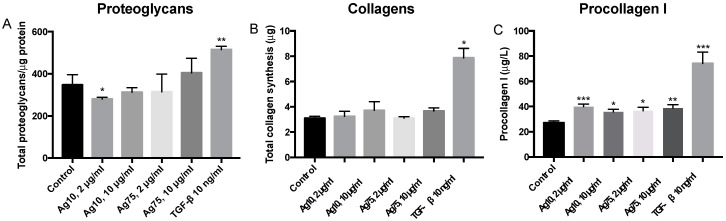
Effect of AgNPs and the positive control transforming growth factor (TGF-β1) on synthesis of (**A**) proteoglycans (n = 4), (**B**) total amount of collagens (n = 4) and (**C**) procollagen I (n = 12) in HFL-1 compared to control at 72 h of exposure. Data are presented as mean +/− SEM, and statistical analyses are compared to untreated controls. * *p* < 0.05, ** *p* < 0.01, *** *p* < 0.001.

**Figure 5 nanomaterials-10-01868-f005:**
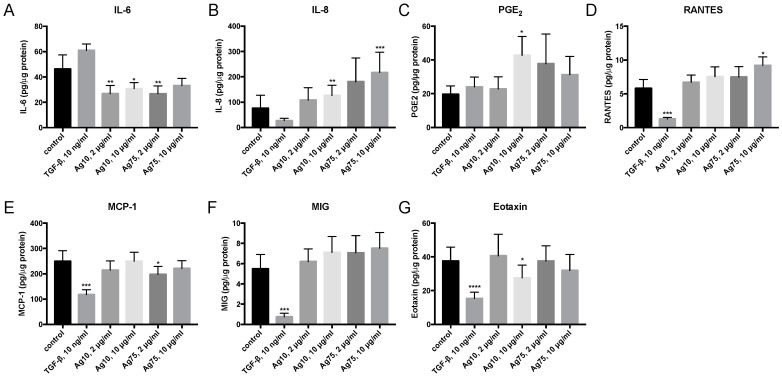
Effect of AgNPs and TGF-β1 10 ng/mL on secretion of the cytokines (**A**) interleukin (IL)-6, (**B**) IL-8, (**C**) prostaglandin E_2_ (PGE_2_), (**D**) regulated upon activation, normal T cell expressed and secreted (RANTES), (**E**) monocyte chemoattractant protein 1 (MCP-1), (**F**) monokine induced by gamma (MIG) and (**G**) eotaxin in HFL-1 (n = 12) after 72 h of exposure. Data are related to the amount of protein in each individual sample and presented as mean +/− SEM. Statistical analyses are compared to untreated controls. * *p* < 0.05, ** *p* < 0.01, *** *p* < 0.001, **** *p* < 0.0001.

**Figure 6 nanomaterials-10-01868-f006:**
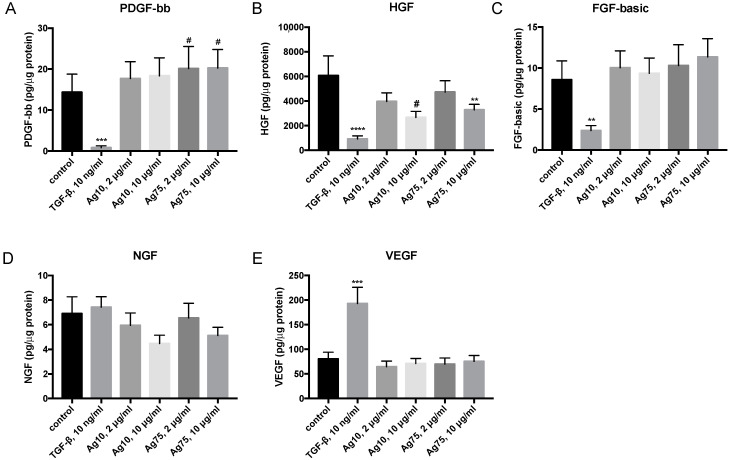
Effect of AgNP and TGF-β1 10 ng/mL on secretion of the growth factors (**A**) platelet derived growth factor (PDGF)-BB, (**B**) human growth factor (HGF), (**C**) fibroblast growth factor (FGF-basic), (**D**) β-nerve growth factor (NGF), and (**E**) vascular endothelial growth factor (VEGF)-A in HFL-1 (n = 12) after 72 h of exposure. Data are related to the amount of protein in each individual sample and presented as mean +/− SEM. Statistical analyses are compared to untreated controls. * *p* < 0.05, ** *p* < 0.01, *** *p* < 0.001, **** *p* < 0.0001, # Trend (*p* = 0.055).

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
