# Peer review of "Silver Nanoparticles Alter Cell Viability Ex Vivo and in Vitro and Induce Proinflammatory Effects in Human Lung Fibroblasts"

_nanomaterials, 2020, doi:10.3390/nano10091868_

Round 1

Reviewer 1 Report

The authors present evidence for for cytotoxicity of small and large AgNPs in a human fibroblast cell line and in precision cut lung slices (PCLS) from mouse lungs after exposures longer than 24 hours. Furthermore they showed localization of AgNPs between alveoli and capillaries in the parenchyma of PCLS and in vaculoes or lysosomal structures in human fibroblasts. They reported mitochondria with altered appearance that might correlate with reduced metabolic activity. Procollagen I expression was increased in fibroblasts and cytokine expression was elevated for IL8, RANTES and MIG as was PGE2. No major effects on growth factors were reported.

Major comments

  1. The growth factor and cytokine measurements were done on human fibroblasts in cell culture; however, a experiments also used PCLS for ex vivo experiments. Did the authors measure cytokines and growth factors in the PCLS samples? In the context of a tissue, this could differ from the observations of the single cell type in culture. They should do if feasible or report if done.
  2. The authors propose that elevated Il-8 expression could lead to increased neutrophil migration. The authors should test this by using a transwell migration assay with human fibroblasts and isolated neutrophils. Do the changes observed in cell culture have a functional impact?
  3. If the authors have access to a Seahorse, then they should measure the effect of the AgNPs on mitochondrial function. This is an interesting finding.

Minor comments

A discussion of whether the doses tested in this work would be achieved in vivo is necessary for understanding the biological significance of this work. Would one be exposed in the lung to the levels used in cell culture?

Reviewer 2 Report

This paper presents the evaluation of toxicity of two sizes of AgNPs 10nm and 75 nm and compares toxicity in ex vivo PCLS from mouse and in human fibroblasts. The authors demonstrate that it is important to look at toxicity at longer time points beyond 24h than previously evaluated by other work, as they observed greater toxicity beyond this time point not observed by others. The paper presents an important topic, especially due to the prevalence of AgNPs in various applications. However, several revisions are needed to improve the strength of this work and the ability to compare these results with other work, as indicated below.

  1. The manuscript would benefit from significant language editing, particularly in the beginning abstract, introduction and discussion sections. Several grammatical/wording issues are present and distract from the work. Furthermore, please write out abbreviations especially for NGF, HGF etc even though these are common for the field, they are less known to the broader audience.
  2. The authors compare viability of PCLS and HFL-1 cells in response to the AgNPs using two different methods (WST-1 for PCLS and HFL-1, and LDH for HFL-1). Since these assays measure different aspects of toxicity (proliferation vs metabolism) It would be useful to also analyze PCLS by LDH. Furthermore, the authors do not explain the differences in the pattern observed for HFL-1 compared between LDH and WST-1 methods. For example, at 48h of treatment, in figure 1D, Ag10 2ug/mL treatment induces more metabolic activity, but induces more cytotoxicity in Figure 2B. Similarly, Ag75 2ug/mL has comparable metabolic activity as the control in figure 1D but induced more cytotoxicity in Figure 2B.  Including a positive control is necessary for the PCLS WST-1 assay, as it is unclear whether these kinds of changes in metabolic activity of only 0.05 Abs/mg, while possibly statistically significant, are actually biologically significant. Furthermore, please include the Ag75 10 ug/mL data in figure 2B. 

It seems 20% cytotoxicity for the control cells is quite high (figure 2B) which questions the health of these cells and needs to be repeated or discussed.

Minor: changing figure 2A and 2B into %viability instead of %cytotoxicity would make it easier to directly compare this to Figure 1 results in which you have a positive value for metabolic activity.

III. Furthermore, the authors point out inconsistencies between their findings and those of others. However, these references use different sources of cells to demonstrate lack of toxicity (normal human dermal fibroblasts) (ref 23) versus the lung fibroblasts used here (HFL-1).  Also, the PCLS from this study were obtained from mouse lung, whereas the study the authors are comparing to used rat lung (ref 24).  Authors point out the difference in localization of the AgNPs in the rat lung PCLS in ref 24 compared to the localization in their study which may also be a factor of the differences in tissue between the two species. While performing PCLS studies with rat lung would be significant effort, it would be important to compare the results with the author’s AgNPs in normal human dermal fibroblasts, as these are readily obtained commercially such that a true comparison can be made with these studies.

Reviewer 3 Report

The manuscript provided by Löfdahl et al. investigated the effect of silver nanoparticles (AgNP) on metabolic activity, cytotoxicity, cell number, ECM dynamics, cytokine and growth factors dynamics, as well as how the particles distribute within the cells and ex vivo tissues. The paper overall was well written, however a few typos and grammatical errors are present. These minor issues do not detract from the manuscript. The studies were well designed and used a novel ex vivo lung assay that provided additional insight into the studies. While the manuscript as written is strong, a few minor points should be changed before publication.

Figure 1C and 1D should be normalized, it can be challenging to properly interpret the results without properly normalization. 

How was the cell cytotoxicity data normalized?  It looks like the cytotoxicity increases in the control after 48 h, and it isn't clear why. Is this a product of the method used for normalization of the data? The cytotoxicity in the control is higher than one would expect, why is the cytotoxicity of the cells between 10-20%?  Is this a function of the assay sensitivity?

It would also be helpful if the cell amount in Figure 2C and D were normalized.  It would also be good for Figure 3A and B to be the same scale, it is hard to compare the size of the particles given the two different magnifications used. In figure 3C D and E please include arrows to indicate the areas of interest you mention in lines 198-202.

While outside of the scope of this manuscript, it may be interesting to see if the cells are simply not excreting the collagens.  The cells may be making elevated levels of the proteoglycans and collagens, but may be limited in their ability to secrete these proteins in response to the nanoparticles. 

For the cytokine assays did the measurements account for differences in cell number? It isn't clear to me that it was accounted for, which would significantly effect the findings. 

What were the limitations of this particular study?  A section on the study limitations would be good thing to include.

Round 2

Reviewer 1 Report

The authors addressed the comments of the reviewers and acknowledged that additional functional studies are needed and described in vivo relevance and limitations of the current study.

Reviewer 2 Report

The authors have improved the language significantly and included additional data. While it would make the message of the paper more significant to have one cell type or PCLS comparison that is the exact same as other studies as mentioned in the previous review, the current changes are sufficient.